# MOCCASIN: a method for correcting for known and unknown confounders in RNA splicing analysis

Barry Slaff[1,8], Caleb M. Radens[2,3,8], Paul Jewell[2], Anupama Jha[1], Nicholas F. Lahens [4], Gregory R. Grant[2,4], Andrei Thomas-Tikhonenko[5,6], Kristen W. Lynch[2,3,7] & Yoseph Barash [1,2,3 ✉]

The effects of confounding factors on gene expression analysis have been extensively studied following the introduction of high-throughput microarrays and subsequently RNA sequencing. In contrast, there is a lack of equivalent analysis and tools for RNA splicing. Here we first assess the effect of confounders on both expression and splicing quantifications in two large public RNA-Seq datasets (TARGET, ENCODE). We show quantification of splicing variations are affected at least as much as those of gene expression, revealing unwanted sources of variations in both datasets. Next, we develop MOCCASIN, a method to correct the effect of both known and unknown confounders on RNA splicing quantification and demonstrate MOCCASIN's effectiveness on both synthetic and real data. Code, synthetic and corrected datasets are all made available as resources.

[1] Department of Computer and Information Sciences, School of Engineering, University of Pennsylvania, Philadelphia, PA, USA. [2] Department of Genetics, Perelman School of Medicine, University of Pennsylvania, Philadelphia, PA, USA. [3] Graduate Group in Cell and Molecular Biology, Perelman School of Medicine, University of Pennsylvania, Philadelphia, PA, USA. [4] Institute for Translational Medicine and Therapeutics, Perelman School of Medicine, University of Pennsylvania, Philadelphia, PA, USA. [5] Department of Pathology and Laboratory Medicine, Perelman School of Medicine, University of Pennsylvania, Philadelphia, PA, USA. [6] Division of Cancer Pathobiology, Children's Hospital of Philadelphia, Philadelphia, PA, USA. [7] Department of Biochemistry and Biophysics, Perelman School of Medicine, University of Pennsylvania, Philadelphia, PA, USA. [8] These authors contributed equally: Barry Slaff, Caleb M. Radens. ✉email: yosephb@upenn.edu

RNA-Seq is an experimental technique that quantifies the relative abundance of RNA molecules in a sample through sequencing of short RNA fragments. By mapping those sequences to a transcriptome or an annotated genome, researchers can quantify expression levels and alternative splicing of genes. RNA-Seq is commonly used in a variety of analysis tasks like the identification of differentially expressed genes or isoforms between two or more conditions; the identification of genes and/or samples that cluster together by gene expression or RNA splicing variations; and quantitative trait loci analysis to identify genetic variants associated with changes in expression (eQTL) or splicing (sQTL).

The results of such analyses can be greatly affected by unwanted factors such as sequencing lane[1,2] or processing batch[3,4]. These factors are either already known prior to starting the analysis task (e.g., sequencing lane), or unknown (e.g., difference in mouse diet which was never recorded). Generally, such confounders (also termed nuisance variables) affect the variability of the data and, depending on the relationship to the biological signal of interest, can lead to inflated rates of false positives and false negatives. Fortunately, for gene expression analysis, there are well-established methods to remove or account for both *known* and unknown confounding factors, such as limma[5], ComBat[6], RUV[7,8], and surrogate variable analysis (svaseq)[9,10]. All of these tools have been applied in many studies and are highly cited.

For the quantification of alternative splicing (AS), a variety of methods and approaches exist. Some methods quantify whole transcripts, usually by assuming a known transcriptome. Many methods, such as RSEM[11], SALMON[12], and Kallisto[13] estimate some form of normalized transcript expression (TE). These estimates can then be corrected for confounder effects using the methods listed above. Other methods, such as MISO[14] and BANDITS[15], estimate relative transcript usage (TU), i.e., the fraction of transcripts from each gene's isoform. In contrast, many other methods focus on "local" splicing changes, quantifying expression at the exon levels (e.g., DEXSeq[16]), or the relative usage of specific RNA segments or splice junctions. This latter approach, described in more details below, involves quantifying the percent spliced in (PSI) for local splice variations (LSVs), or AS events. AS events commonly captured by these methods include exon skipping (where PSI captures the fraction of isoforms including or excluding an exon) and alternative 3'/5' splice sites. Notably, PSI-based quantification of AS events has been the main approach in the RNA splicing field and is also the focus of the work presented here. Some of the reasons for this focus on PSI quantification in the RNA splicing field include the ability to more accurately and easily quantify local AS events from junction spanning RNA sequencing reads, the ability to capture complex and un-annotated (de-novo) splice variants which are particularly relevant for disease studies, and the ability to validate and manipulate such AS events using RT-PCR and mini-gene reporter assays[17].

However, in contrast to the common usage of PSI quantification methods for the study of RNA splicing, there is a clear lack of tools for modeling known and unknown confounding factors in PSI-based splicing analysis. We suspect this scarcity of tools reflects a general lack of awareness of the effect of confounders on RNA splicing quantifications. Specifically, we were not able to find any previous work that quantitatively assessed the effect of confounders on splicing analysis and compared it to the effect on gene expression.

One challenge with correcting for confounders in RNA splicing analysis based on PSI estimates, is that the tools designed for correcting expression estimates are not well suited for this task. Local splicing variations (LSV) are typically quantified from junction reads, a subset of exon-mapped reads spanning across introns. When junction reads from an exon map to two or more alternative splice sites up/downstream, the percent splice inclusion (PSI) of each of these splice sites is quantified as the ratio of junction reads mapping to that splice site over the total number of junctions reads mapped to all splice sites at that locus. Similarly, between samples or groups of samples, the delta PSI (dPSI) quantifies the extent of changes in PSI, or differential splicing. Consequently, alternative splicing (AS) quantification is distinct from that of gene or transcript expression: rather than real values or log fold change, PSI values are in the range of 0 to 1, or −1 to +1 for dPSI. LSV PSI quantification also suffers from different biases and challenges than those typical in expression estimates. For example, in typical RNA-Seq experiments many LSVs suffer from low coverage in terms of junction spanning reads, which in turn limits the ability to quantify changes in PSI associated with those. On the other hand, issues such as effective transcript length or 3' bias in read coverage do not play a major role in PSI quantification as those can benefit from the built in normalization between the reads associated with an LSV's junctions.

Most previous work involving PSI-based splicing analysis and confounding factors utilized pipelines originally built for QTL analysis. For example, Raj et al.[18] quantified variations in intron cluster usage using LeafCutter[19] and then applied fastQTL[20] to the data matrix. This analysis implicitly assumes gaussian distributions, standardizes the data, and then corrects for confounders such as age or gender. However, not only do PSI values not follow a Gaussian distribution, but the corrected values are commonly negative or greater than one, losing their interpretation as splicing fractions. Furthermore, these pipelines use only point estimates for PSI, losing read coverage information which is crucial for controlling false positives in differential splicing analysis[21]. Finally, while some software allow users to specify confounders for differential splicing analysis[16,19,21,22], we are not aware of any tool that is able to correct for both known and unknown confounders, outputting corrected junction spanning reads for both supervised (e.g., differential splicing) and unsupervised (e.g., clustering) splicing analysis tasks.

In order to address the lack of tools for correcting confounders in RNA-Seq based splicing analysis we developed MOCCASIN (Modeling Confounding Factors Affecting Splicing Quantification) algorithm, which operates by jointly adjusting the estimated read-rate for the junctions in each LSV. At its core, the MOCCASIN model is notably simple: read rates across samples are all scaled to the median of the total read rates across all junctions in the matching LSV. As a result, adjusting a junction's read rates across samples adjusts its respective PSI across samples, up to a scaling factor. Optionally, read rates can be log scaled to capture log linear effects which have theoretical support[23] though we show that in MOCCASIN's model formulation a simple linear model performs well. Next, each junction's read rates are fit, using vanilla ordinary least squares regression (OLSR) to a linear combination of factors, which may include both known and unknown (learned) factors. The unwanted factors of variations are then removed, leaving corrected read rate per junction which are then renormalized and scaled back to the original coverage level.

The simplicity of the MOCCASIN model allows it to be highly efficient in both memory and compute time, while still being able to effectively correct the effect of confounders on splicing quantification from RNA-Seq experiments. MOCCASIN is implemented in python as an open source package. MOCCASIN takes advantage of specific features characteristic to MAJIQ's[24] LSV quantification (see Methods), but it also implements a general API so other PSI quantification algorithms can easily employ it. We evaluate MOCCASIN's performance on both synthetic and real datasets under a variety of settings, demonstrating that it

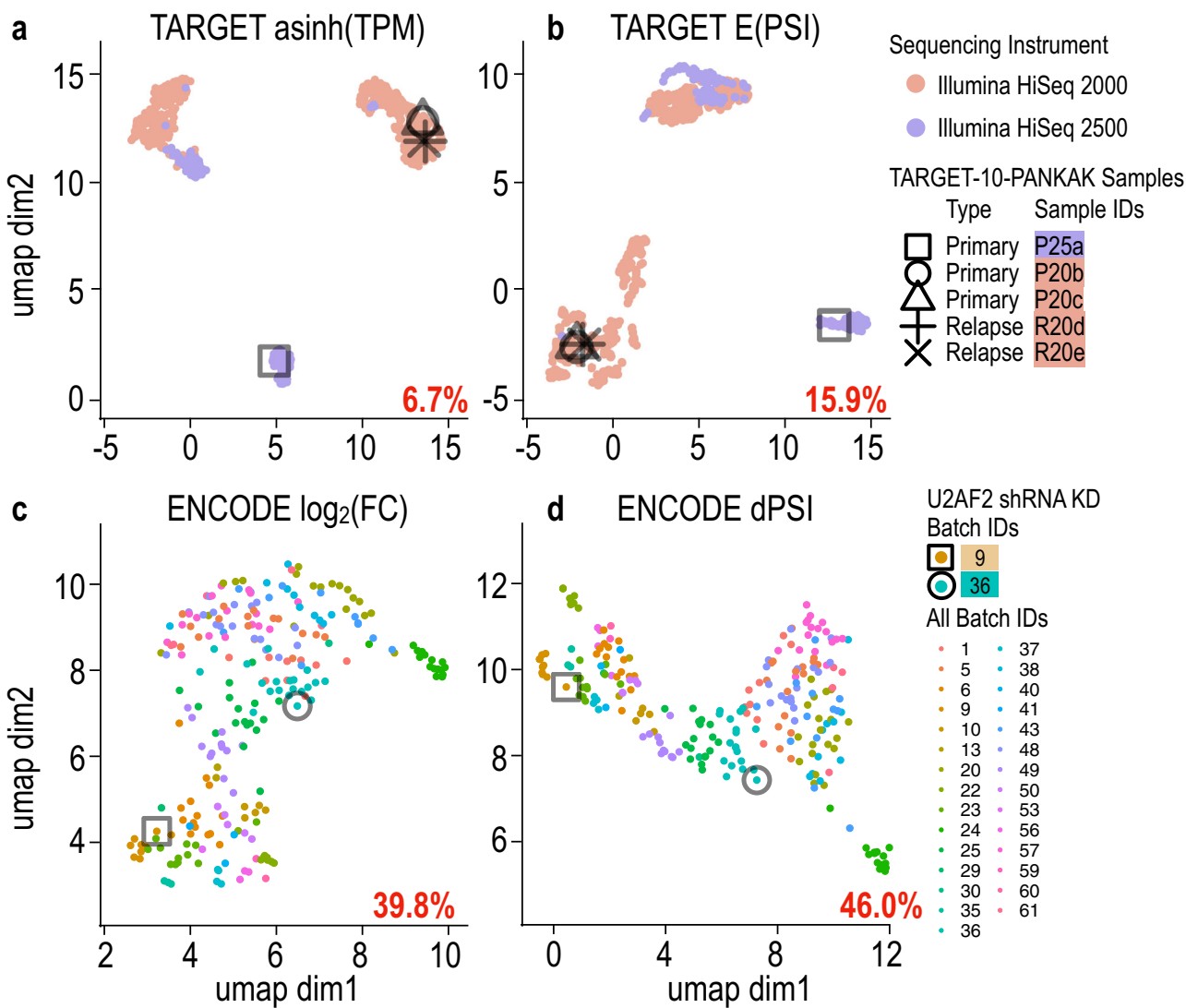

**Fig. 1 Batch effects impact both gene expression and splicing analysis.** Uniform manifold approximation and projection (UMAP) of gene expression analyses (**a**, **c**) and splicing analysis (**b**, **d**) for TARGET (top, $N = 870$) and ENCODE (bottom, $N = 489$). Colors indicate batch identity. Numbers in red represent percent of total variation ($R^2$) associated with batch in each dataset. Shapes mark samples from the same patient (TARGET, patient TARGET-10-PANKAK) or experiment type (ENCODE, U2AF2 KD) which cluster by batch. TPM, transcripts per million; FC, Fold change; dPSI, delta percent splice inclusion.

works effectively to eliminate false positives and recover true biological signals diminished due to confounders.

## Results

**Confounders can have a strong effect on RNA splicing analysis.** First, we set out to assess the effect of confounders on both expression and splicing analysis in two large and highly used datasets, TARGET (Therapeutically Applicable Research to Generate Effective Treatments initiative) and ENCODE[25], each with hundreds of samples. The ENCODE dataset comprises 236 and 238 shRNA knockdown experiments performed in K562 and HepG2 cell lines, respectively, with many of the knockdown experiments targeting RNA binding proteins. The ENCODE experiments were done in 61 distinct batches, encompassing a total of 122 control and 978 knockdown RNA-Seq samples. The TARGET dataset comprises samples from 250 and 329 pediatric patients with B and T Cell of origin acute lymphoblastic leukemias (ALL), respectively. Samples were taken at primary diagnosis and/or after relapse and sequenced in multiple batches. See Supplementary information for more details about both datasets.

As shown in Fig. 1 we found the effect of confounders on splicing analysis is at least as large as the effect on expression analysis. Specifically, in the TARGET's leukemia dataset and ENCODE's shRNA knockdown (KD) dataset the batch labels were associated with 15.9% and 46%, respectively of the total splicing variations as measured by $R^2$ (see "Methods") compared to 6.7% and 39.8% for expression. Indeed, Van Nostrand et al. noted such batch effects for ENCODE[26], but this effect was not quantified and we are unaware of a previous publication reporting batch effect associated with the sequencing machine for the TARGET dataset. Importantly, batch effects are not restricted to such large consortium data. We observed similar results for mouse tissue samples previously used to study expression batch effects by Peixoto et al[4]. (See Supplementary information and Supplementary Fig. 1).

**MOCCASIN effectively corrects the effect of known and unknown confounders in RNA splicing analysis.** In order to address the need for correcting confounding factors in splicing analysis we developed MOCCASIN. Briefly, MOCCASIN input

consists of two elements. One is a matrix $R$ with estimated read rates $R_{mk}$ for each junction $m \in [1 \dots M]$ in each sample $k \in [1 \dots K]$. For many PSI quantification algorithms $R_{mk}$ is simply the total junction spanning reads in experiment $k$ that support junction $m$. The junctions represented in the matrix $R$ are assumed to be grouped into LSVs, with a separate data structure given as input and identifying each junction row $m$ with its matching LSV $l$. The second input element for MOCCASIN is a design matrix for the RNA-Seq experiments, allowing the user to specify a set of variables of unwanted variations (e.g., batch ID) and variables of interest (e.g., tissue type). In addition, MOC-CASIN also allows users to specify a number of unknown factors of variations to be learned from the data using a factor analysis approach similar to that used by RUV[7] (see "Methods"). The set of read rates $R_{mk}$ are scaled across all samples; fit to the variables of wanted and unwanted variations using OLSR; adjusted to remove the unwanted sources of variations; then scaled back to the original total read rate per LSV in each sample such that MOCCASIN outputs a "cleaned" read rates matrix (see methods for more details). The resulting corrected read rates matrix can then be fed into any downstream algorithm involving tasks such as PSI quantification for clustering analysis or detection of differentially spliced events. Specifically, MOCCASIN was designed to work in conjunction with MAJIQ but other algorithms for PSI or dPSI quantifications can also be used via MOCCASIN's API.

To assess the effectiveness of MOCCASIN we first created a "realistic" synthetic data, using BEERS[27] to simulate a total of 16 real samples of RNA-Seq from mouse tissues[28] (Aorta and Cerebellum). Simulating real samples allowed us to capture realistic expression levels and variability between samples in and between the original tissue groups (see "Methods"). After the original samples were simulated, we introduced a synthetic batch effect to half of the samples denoted as batch B. Specifically, $G \in 2, 5, 20$ percent of randomly chosen genes in batch B were perturbed by reducing the TPM of their most highly expressed isoform by $C \in 2, 10, 60$ percent. Next, we randomly selected another of the gene's isoforms and increased its expression by the same amount, thus maintaining the gene's overall original TPM (see methods for additional details on the simulation process). Finally, to avoid introducing additional sources of variations, quantifications of PSI and dPSI were performed with the same settings of MAJIQ before and after applying MOCCASIN to assess its effectiveness for batch correction.

Figure 2 summarizes the evaluations of MOCCASIN on the above synthetic data. Figure 2a shows the cumulative distribution of the changes in PSI per LSV due to the injected batch effect ($G = 20\%$ and $C = 60\%$) compared to the ground truth unperturbed data, either before MOCCASIN (blue), or after MOCCASIN where we vary the total number of input samples from 4 (purple) to 8 (brown), 12 (yellow), or 16 (orange). For example, we see that the number of LSVs with dPSI > 0.2 drops from 1229, (~6% of total LSVs, blue line) to 345-151 (~1–2%, purple, brown, yellow, and orange lines for 1, 2, 3, or 4 samples per batch and tissue combination) i.e., up to 88% reduction in the number of highly perturbed LSVs.

While the overall accuracy in PSI correction as captured by Fig. 2a is important, it still leaves open the question of how does the batch correction affect the accuracy of detecting differentially spliced LSV for the signal of interest. To address this question, Fig. 2b shows the batch effect on the number of detected differentially spliced LSV ($y$-axis) as a function of the difference's significance ($x$-axis, Student's $t$ test $p$-value, see "Methods"). Here, with $G = 20\%$ and $C = 60\%$, we see the batch injected data (blue) compared to the "ground truth" data before batch injection (green) causes inflation of both batch-associated differences (Fig. 2b), as well as false positives for the tissue signal (Fig. 2c).

Applying MOCCASIN (orange or gray) effectively controls both the batch signal (Fig. 2b) and the false positives for the biological signal (Fig. 2c) with as few as 2 input samples per tissue/batch (see Supplementary Fig. 2) with only a moderate loss of true biological signal (false negative) due to over correction at the moderate $p$-value levels. To better quantify these effects across a range of settings we applied MAJIQ's Bayesian dPSI model to report all LSVs for which the posterior probability of a PSI change of above 20% is at least 95% (P(|dPSI|> = 20%) > = 95%). This threshold is commonly used to detect splicing changes that are considered biologically significant, of high confidence, and can be validated using orthogonal approaches such as RT-PCR. Figure 2d, e show that small perturbations of $C = 2\%$ or even $C = 10\%$ have little effect on the false positive rate (FPR) with respect to the original batch signal (Fig. 2d) and the false discovery rate (FDR) with respect to the tissue signal (Fig. 2e). These results are to be expected given the dPSI thresholds set for MAJIQ, reflecting its robustness to small perturbations below the specified detection threshold. However, the picture changes dramatically when the injected signal reaches $C = 60\%$, with both FPR for batch signal and FDR for tissue signal climbing proportionally as $G$ increases from 2% to 5% and 20% (blue). Subsequently, applying MOCCASIN (orange) effectively controls for the batch effect and perhaps most importantly maintains an empirical FDR of 6–8%, close to the Bayesian estimate of 95% confidence. We also found that using known batch covariates or treating those as unknown and letting MOCCASIN discover and correct them performed similarly (gray), and this result was not sensitive to the number of unknown confounders used (compare U1, U2, U3 in Supplementary Fig. 3). See Supplemental Information for additional plots for other statistics and other settings for batch effects.

As the last component of evaluations using synthetic data, we assessed the effect of MOCCASIN's batch correction on down-stream unsupervised clustering. Figure 2f shows that the original data clusters by tissues (left), by batch when the batch signal was injected (middle, $G = 20$ and $C = 60$), and again by tissue after MOCCASIN is applied (right). Similar results are observed when MOCCASIN is configured to identify an unknown confounder (Supplementary Fig. 4).

**MOCCASIN boosts biological signals in TARGET and ENCODE datasets.** Next, we applied MOCCASIN to the TAR-GET and ENCODE datasets described above, each of those representing highly used datasets with a different set of challenges for confounders correction. In the case of ENCODE, there is a high number of batches (61), and each batch consists of RNA-Seq experiments in one of two cell lines (HepG2 and K562). Every KD experiment had 2 biological replicates within the same batch, and each batch had its own two biological replicate controls (the matching cell line without any KD perturbation). After some experimentation we converged to a procedure where MOCCA-SIN was run separately for each cell type, removing variation due to the 29 and 32 batch identifiers in the HepG2 and K562 samples, respectively. The result of this correction procedure was that variance explained by batch dropped from 46% as shown in Fig. 1d to 4.3% as shown in Fig. 3a, whereby a UMAP analysis shows no batch-associated clustering of samples. Next, taking advantage of the fact U2AF2 experiments were repeated in two batches (9 and 36), we contrasted the differential splicing detected in those batches before and after MOCCASIN's correction. We identified 2595 and 1897 U2AF2 KD-induced changes in splicing (|dPSI|> = 0.1, $p$-value < 0.05) in batches 9 and 36, respectively, of which only 773 were identified in both batches (Fig. 3b, left). After applying MOCCASIN (Fig. 3b, right)

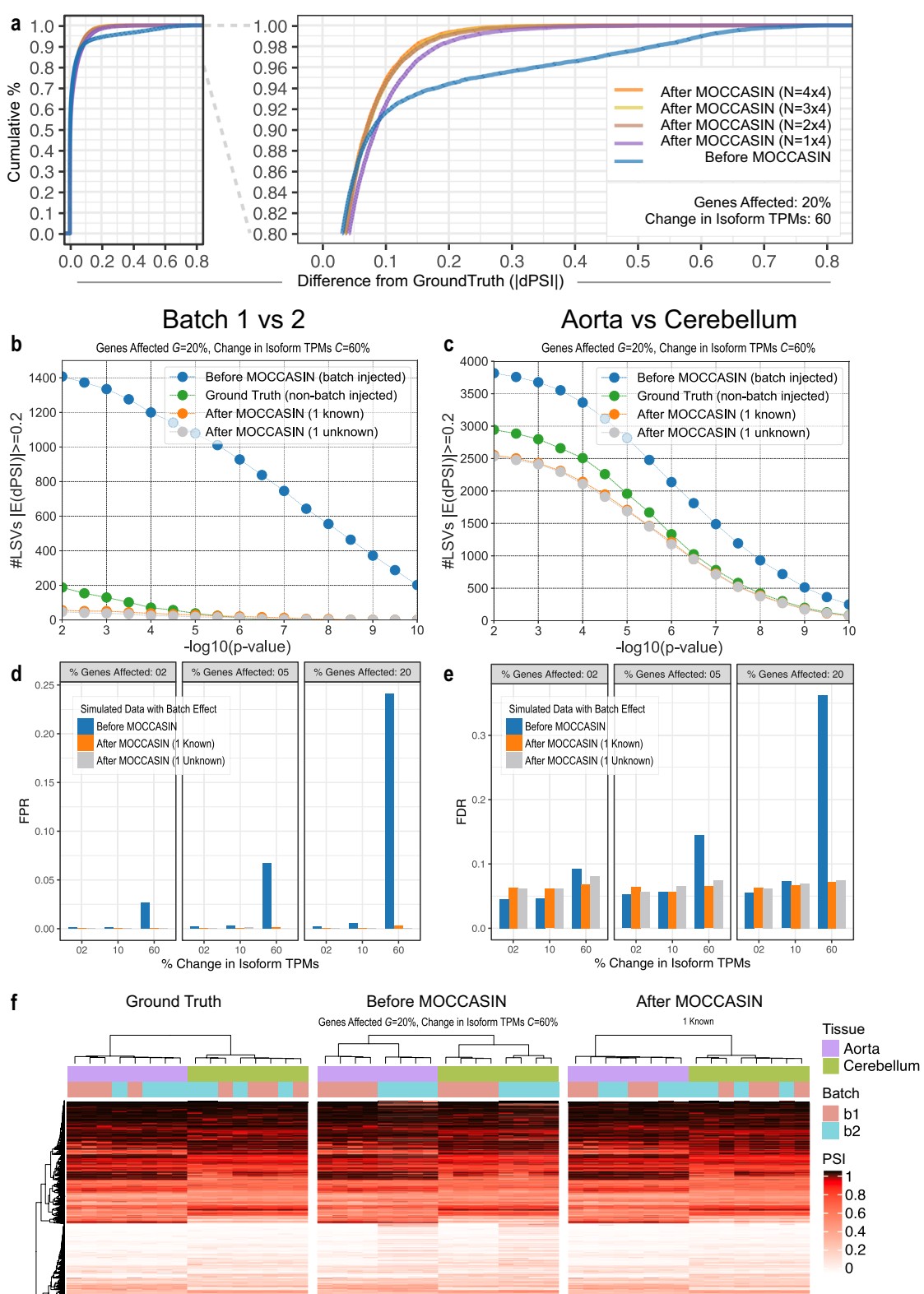

to model and remove batch-associated variation, the overlap between differentially spliced events upon KD of U2AF2 in batch 9 and 36 increased from 773 to 1048, the overall Pearson correlation of dPSI between batches increased from 0.71 to 0.79, and the total number of detected U2AF2 KD-induced changes increased by 12.2% and 8.1% for batches 9 and 36, respectively.

The TARGET dataset introduced another set of challenges for correcting the effect of confounding variables on PSI quantifications. First, we identified a strong batch effect associated with the sequencing platform (HiSeq 2000 vs 2500), which to the best of our knowledge was not reported before. Our initial analysis was successful in controlling for the unwanted variation associated

**Fig. 2 Removal of batch effects from simulated RNA-Seq data with MOCCASIN.** RNA-Seq samples from mouse Aorta and Cerebellum were simulated using BEERS while injecting $G\%$ of the genes in half the samples with a batch effect of $C\%$ expression change of the main isoform (see main text). **a** Cumulative distribution of the difference ($|dPSI|$) from simulated ground truth after batch signal injection ($G = 20\%$, $C = 60\%$) either before MOCCASIN (blue) or after correction with increasing numbers of samples for each of the four batch/condition combinations: $1 \times 4$ (4 total, purple), $2 \times 4$ (8 total, brown), $3 \times 4$ (12 total, yellow), and $4 \times 4$ (16 total, orange). All plots are derived from the same representative sample (SRR1158528) to maintain a fixed base for comparison, with similar plots observed for other perturbed samples (data not shown). Total number of LSVs: 21566. **b**, **c** The number of LSVs ($Y$-axis) detected as differential ($|E(dPSI)| > 0.2$) for the batch 1 ($N = 4$) versus batch 2 ($N = 4$) signal (**b**, left) and the aorta ($N = 4$) vs cerebellum ($N = 4$) signal (**c**, right) across a range of increasingly significant $p$-values ($X$-axis, Student's $t$ test, $-\log_{10}$ scale). Number of samples used is 4 per batch/tissue combination (same as in the orange line in **a**). The green points ("Ground Truth") are from the simulated data with no batch signal injection and the blue points ("Before MOCCASIN") are from the same data after batch signal injection ($G = 20\%$, $C = 60\%$). Both blue and green points serve as reference points for MOCCASIN correction of the batch signal. Orange and gray represent, respectively, the results after MOCCASIN correction when the batches are known or unknown. **d**, **e** Assessing false positive rate (FPR) for the batch signal (d, left) and false discovery rate (FDR) for the tissue signal (e, right) for a range of $G$ values (2, 5, 20%) and $C$ effect size (2, 10, 60%). Number of samples same as in (**b**, **c**). Here positive events where considered as those changing by at least 20% with high confidence by MAJIQ ($P(|dPSI| > 0.2) > 0.95$). Under these definitions small effect sizes ($C = 2,10\%$) represent perturbations that are not expected to affect the positive event set much. **f** Heatmaps of E(PSI) from simulated data without batch effect (ground truth, left), with simulated batch effect ($G = 20\%$, $C = 60\%$) without correction (middle), and after applying MOCCASIN with 1 known confounding factor (right). Each column is a sample, and each row is an LSV ($N = 4941$). The colored bars above the samples denote the sample's tissue (8 aorta samples in purple, and 8 cerebellum samples in green) and batch (8 batch 1 sample in red and 8 batch 2 samples blue).

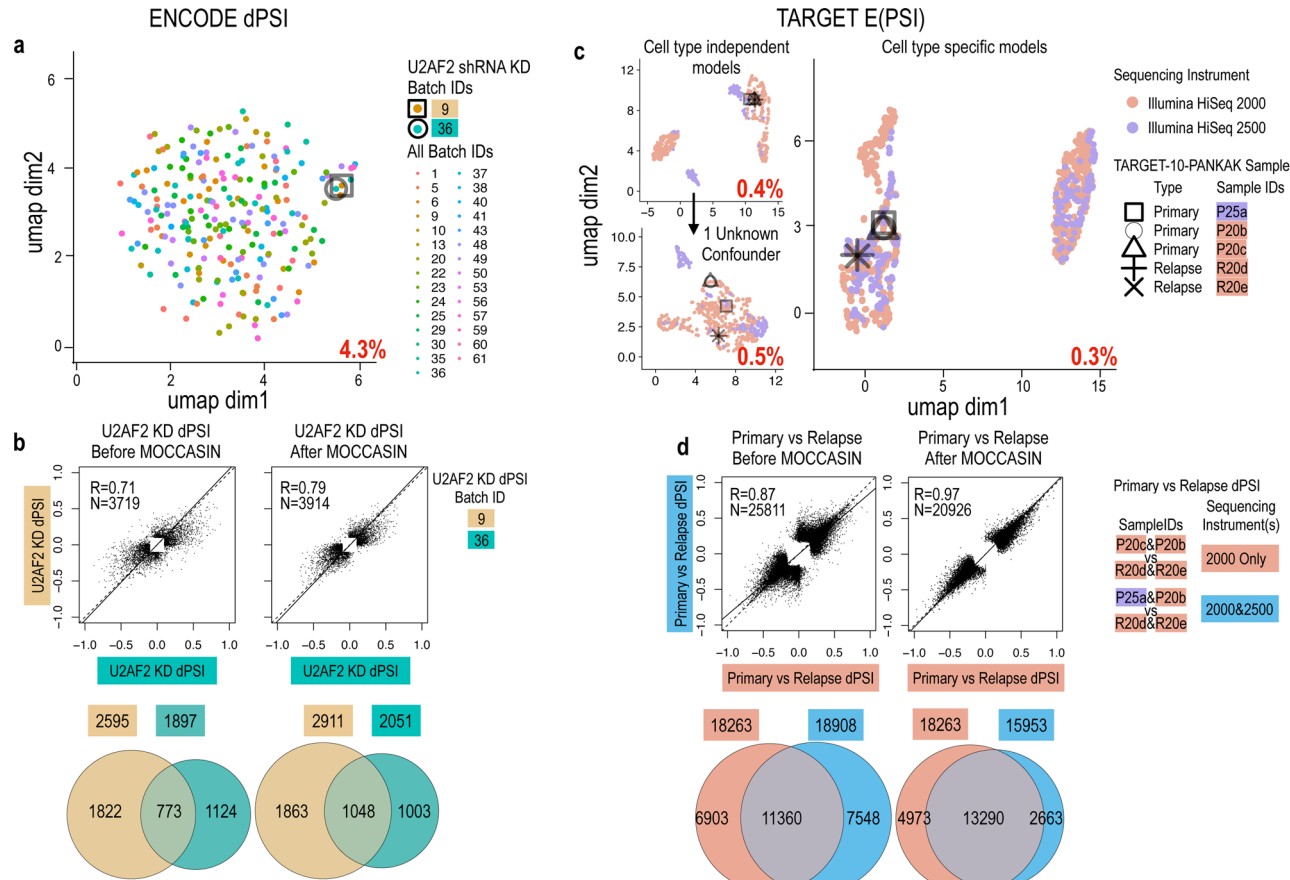

**Fig. 3 Batch correction of TARGET and ENCODE datasets. a** UMAP plot for ENCODE ($N = 489$) as in Fig. 1 but after applying MOCCASIN. U2AF2 knockdowns (square and circle) now cluster together, percent of total variance ($R^2$) associated with batch drops to 4.3% (red text) **b** Pearson correlation of significant splicing changes upon U2AF2 KD (dPSI > 0.2) between batches increases from $R = 0.71$ (top left) to $R = 0.79$ after MOCCASIN (top right). Similarly, the number of significantly changing events ($P(|dPSI| > 0.2) > 0.95$) in each batch and the overlap of these events between batches increases after MOCCASIN correction (**c**, compare bottom left and right). **c** UMAP plot for TARGET ($N = 870$) as in Fig. 1 but after applying MOCCASIN without specifying a cell type (top left), after adding an unknown confounder (bottom left), and when using a cell type specific model after inferring missing cell type labels (right). TARGET technical replicates P25a-c cluster together (square, circle, and triangle), and $R^2$ drops to 0.3% (red text). **d** Pearson correlation of significant splicing changes between primary and relapse samples from the same patient (TARGET-10-PANKAK) increases from $R = 0.87$ (top left) to $R = 0.97$ (top right) after correcting for the sequencing platform. Note that since only the HiSeq2500 samples are corrected by MOCCASIN, only the set that includes those (cyan) is affected. Accordingly, the total number of significantly changing events ($P(|dPSI| > 0.2) > 0.95$) drops for the cyan set from 18908 (bottom left) to 16100 (bottom right) but the overlap with the other set increases (11360 to 13290).

with the sequencing platform, dropping the amount of variance explained by it, from 15.9% to 0.4% (compare Fig. 1b to Fig. 3c). However, clear structure was still evident in the data. Initially, we were able to successfully remove that structure by adding one unknown confounder to the MOCCASIN model (Fig. 3c, bottom left plot). However, the amount of variance associated with the unknown confounder (9.8%) motivated us to search for a biological explanation for it. Indeed, we found that many (but not all) samples were annotated by cell type, a label we originally missed. We then inferred the missing cell type labels based on the initial embedding and performed cell type specific corrections, yielding the corrected data UMAP shown in Fig. 3c (right plot). Finally, we assessed the effect of MOCCASIN on known biological signals in TARGET by comparing a within-batch analysis (same sequencer) to a mixed-batch (multiple sequencers) before and after correction. We found that splicing variations in TARGET data corrected by MOCCASIN correlated better (Fig. 3d, top: Pearson correlations increased from 0.87 to 0.97) and overlapped more (Fig. 3d, bottom: overlap increased by 17%) when detecting patient-specific primary ALL diagnosis versus relapse-associated splicing differences $(P(|E(dPSI)|> = 0.1)> = 0.95)$. In summary, our analysis of two large public datasets demonstrate both the severity of confounder effects on splicing analysis as well as the ability to effectively correct those using MOCCASIN while handling different sources of variations and experimental designs.

## Discussion

While the issue of confounding factors' effect on gene expression analysis has received much attention, the equivalent effect of confounders on RNA splicing analysis has not been well studied. Here we use two large public datasets to show the magnitude of confounders' effect on RNA splicing quantification can be at least as big as that observed for gene expression analysis. We develop MOCCASIN, a dedicated tool to correct for both known and unknown confounders in RNA splicing PSI-based analysis, and demonstrate its effectiveness on both synthetic and real data. We have also made all code and data available. Specifically, in addition to correcting ENCODE HepG2 data (Fig. 3), we also corrected the K562 data (Supplementary Fig. 5). The corrected HepG2 and K562 ENCODE dataset (See Supplementary Information) should serve as a highly valuable resource as it offers quantification of both complex and de-novo splicing variations which were not available before, as well as MOCCASIN batch corrections. Similarly, the synthetic data generated for this study (see Supplementary Methods) can serve as a benchmark for future tool development.

While we focused here on large public datasets it is important to note similar observations can be made in much smaller datasets. For example, the data used in the development of RUV for expression correction[10], exhibited strong batch effects in splicing analysis as well (see Supplementary Fig. 1). Another important point to make is with respect to scalability. Nowadays datasets can easily involve hundreds or even thousands of samples, requiring efficient algorithms. This problem is further compounded by the fact that the number of junctions in the human transcriptome is typically an order of magnitude larger than the number of genes. Consequently, we chose to utilize a relatively simple model which appears to perform well while still scaling to large datasets as those used here. For example, processing the TARGET dataset consisting of 885 samples with 40 threads and 30 GB of RAM took us 67 h with much of that execution time spent on I/O and not the actual MOCCASIN correction algorithm (See Supplementary Fig. 8 and the Supplementary information for further details).

While the simplicity of MOCCASIN enables efficient execution, the underlying model has several limitations that should be noted. First, previous work points to nonlinear modeling effects on read rates and consequent PSI[23]. While in our testing a linear model after a smoothed to zero log (STZL) transformation of the read rates did not improve accuracy (Supplementary Figs. 9, 10) it is possible such a transformation, or other nonlinear models would prove advantageous in other settings. Regardless, the basic MOCCASIN model is limited by the fact that the confounders design matrix has to be full rank. This limitation is corroborated by the fact algorithms that quantify PSI for alternative splicing events typically rely on junction spanning reads and can thus be more sensitive to low coverage, leading to missing values in the data matrix. MOCCASIN handles this issue by checking the design matrix is full rank given the observation for each such splicing event, and reports back which event has been corrected. Future work introducing more elaborate nonlinear models, mixed effects models, and models which share information across the corrected LSVs may address some of current modeling limitations.

Beyond the modeling assumptions, there is also inherent limitation in outputting a single "corrected" version of the original RNA-Seq data. Such output by itself does not include information about the credible interval for the underlying estimated parameters for confounders' effect, and consequent possible variability in the "corrected" output. This issue has already been noted for methods correcting gene expression and downstream analysis such as differential expression[29]. In the case of MOCCASIN, the model enables users to specify multiple estimates for the read rates per junction, which results in multiple corrected "versions" of the read rates. This feature enables MOCCASIN to take advantage of MAJIQ's model which outputs multiple estimates for read rates per junction, and then combines those into posterior probabilities for PSI and dPSI estimates. However, while this MAJIQ feature was designed to accommodate for the uncertainty in PSI estimates it does not directly model uncertainty in confounders coefficients, leaving the modeling of such uncertainty as another future direction to improve upon MOCCASIN's model.

Another direction for future investigation is the detailed characterization and synthetic modeling of confounder effects on PSI estimates. For example, in this study we generated synthetic batch effects by reducing the major transcript for a given gene by a certain percentage and increasing another transcript accordingly to maintain the same level of gene expression. This approach was motivated by previous work that showed that changes between cellular conditions usually increase one splice junction in an LSV while reducing another. Also, recent work demonstrated that PSI can be modeled as the result of factors affecting read rates via an exponential function[23]. However, the landscape of confounders' effects on splicing is yet to be explored. Specifically, while this study included large scale quantification of confounders effects on PSI and dPSI estimates, we still lack good characterization of these effects. Do different confounders observed in different studies cause different effects? How are these effects distributed between different types of alternative splicing events? Such detailed characterization will help inform better correction algorithms as well as more realistic simulations.

In conclusion, we believe much future work remains to fully characterise and hopefully better model confounder effects on PSI quantification from RNA-Seq. Nonetheless, we hope that the combined code, data, and analysis we provided here will serve as a valuable resource for the research community and shed much-needed light on the need to control for confounders in RNA splicing analysis.

                    

## Methods

**Simulated data generation from real RNA-Seq samples**. The mouse tissue simulated data are based on the real mouse cerebellum and aorta data from Zhang et al.[28]. We used transcript-level TPM quantifications from these data as the empirical distributions for simulating RNA-Seq reads with the Benchmarker for Evaluating the Effectiveness of RNA-Seq Software (BEERS)[27]. For all BEERS simulations we used the same GENCODE release M21 gene models[30] for transcript quantification, and the sequence from build GRCm38.p6 of the mouse reference genome (downloaded from the GENCODE website). Briefly, we directly converted the table of TPM values for each input sample into a BEERS feature quantification file so we could simulate RNA-Seq data with the same expression distributions as the input samples. Next, we used BEERS to simulate 30 M paired end RNA-Seq reads for each input sample, with uniform coverage across the length of each expressed transcript and no intronic expression. BEERS introduced polymorphisms and errors into the resulting data with the following parameters: substitution frequency = 0.001, indel frequency = 0.001, error rate = 0.005. The BEERS simulator generated FASTQ files containing the simulated read pairs, a listing of the true alignments and source transcript for each read pair, and the true expression counts for each transcript in the simulated sample.

**Simulation of batch effect**. Batch effects were introduced to the transcript level TPMs of four aorta and four cerebellum samples. The first four of the aorta samples and the first four of the cerebellum samples were defined as "batch 1". The last four of the aorta samples and the last four of the cerebellum samples were defined as "batch 2." Batch effect perturbations were only ever introduced to the batch 2 samples and batch effects were restricted to genes with at least two protein coding transcripts and at least one transcript with ≥10 reads per kilobase of transcript length in every sample. The procedure to introduce batch effects is as follows: first, the most abundant protein coding transcript per gene was identified as the transcript having the maximum over all transcripts of the minimum reads per kilobase over all samples. This definition ensures the selected transcript is not zero in any sample. Then, for a given gene a batch effect was introduced by (1) selecting a transcript uniformly at random (excluding the most abundant transcript) and (2) reducing TPM of the most abundant transcript by a factor of "C% Change in Isoform TPM" and correspondingly increasing the TPM of the randomly selected transcript, thus maintaining the overall TPM of the gene and not breaking the definition of TPM (sum of all TPMs in a sample is one million). The "C% Change in Isoform TPM" factors included 2%, 10%, and 60%. In addition to introducing three different levels of percent changes in isoform TPMs, we also varied the percent of genes batch-effected. We introduced batch effects to G = 0%, 2%, 5%, or 20% of all genes with protein coding transcripts.

**MOCCASIN algorithm**. MOCCASIN adjusts read rates representing evidence for RNA splicing events in order to remove confounding variation. The two main inputs provided by the user are the read rates matrix (dependent variables) and the design matrix which lists the confounding factors and covariates. The read rate input data $R$ is a matrix such that each entry $R_{mk}$ represents the read rate for a splice junction $m \in [1 \ldots M]$ in sample $k \in [1 \ldots K]$. In the simplest case these read rates would simply be the number of junction spanning reads supporting that junction in an experiment. The design matrix specifies which experiment k belongs to which independent variable. We break the design matrix into two groups, corresponding to confounders **C** and variables-of-interest/non-confounders **V**. In the case when the known confounders are supplemented with additionally learned factors of unwanted variation (see below) the columns of **C** are partitioned into known **N** and unknown **U** confounders.

MOCCASIN considers the fundamental splicing unit of interest to be a local splicing variation (LSV), where each LSV refers to a collection of splice junctions with either a common source or target exon[17,31]. Hence, the junctions represented in the read rate table are assumed to be grouped into LSVs, with a separate data structure given as input and identifying each junction row $m$ with its matching LSV $S(m) = l \in [1 \ldots L]$. We note though that other definitions of alternative splicing events can be used (e.g., intron clusters as in ref.[19]), under the assumption that the input elements represent read rates of entities (e.g., splice junctions) that belong to groups such that the relative fraction of each such entity (i.e., PSI) need to be adjusted.

For each LSV $l$ and sample $k$, MOCCASIN first computes the total read rates over junctions in the LSV ($T_{l,k} = \Sigma_m R_{m,k} s.t. S(m) = l$). Then the total read rate per LSV in each sample is scaled to the median total read rates for that LSV across all samples, such that the scaled read rates per LSV maintain the same PSI:

$$\hat{T}_l = median_k T_{l,k}$$

$$\widehat{(R_{m,k})} = R_{m,k} \Omega_{l,k}$$

Where $\Omega_{l,k} = \frac{\hat{T}_l}{T_{l,k}}$ is the scaling factor. After scaling, a function $F()$ over the read rate $\widehat{(R_{m,k})}$ is modeled as a linear combination of covariates with homoscedastic noise:

$$F(\widehat{R_{m,k}}) = \alpha_m + N_k \cdot \gamma_m + U_k \cdot \delta_m + V_k \cdot \eta_m + \epsilon_{mk} \qquad (1)$$

Where $\alpha_m$ is the intercept for junction $m$; $\gamma, \delta, \eta$ are the coefficients corresponding to **N**, **U**, **V** respectively; and $\epsilon_{mk} \sim N(0, \sigma^2)$ is assumed to be homoscedastic gaussian noise over the scaled read rates. MOCCASIN includes two implementations for the function F. The simpler default function is unity $F(x) = x$ which translates to a linear model over the scaled read rates. The second is a smoothed towards zero log (STZL) transformation:

$$F(x) = ln(x) s.t. x > 2 \qquad (2)$$

$$F(x) = ax^2 + bx s.t. 0 \le x \le 2$$

$$F^{-1}(x) = e^x s.t. x \ge ln(2)$$

$$F^{-1}(x) = \frac{1}{2a}\left(-b + \sqrt{b^2 + 4ax}\right) s.t. 0 \le x < ln(2)$$

Where:

$$a = \frac{1}{4}(1 - ln(2))$$

$$b = ln(2) - \frac{1}{2}$$

The STZL transformation has some theoretically desirable properties. It allows for a log linear model where the effect of confounder has a multiplicative effect on read rates, inline with recently suggested models for elements affecting PSI[23]. Similar to arcsinh used for gene expression, variance stabilization[32] STZL avoids exploding negative values for low read rates but unlike arcsinh does not cause variable deviation from the theoretically desirable log transformation for high read rates. See Supplementary Fig. 6 for an illustration of STZL and Supplementary Figs. 8 and 9 for a comparison between the default linear model and STZL.

The simple read rate model defined above allows MOCCASIN to derive estimates of $\hat{\alpha}_m, \hat{\gamma}_m, \hat{\delta}_m, \hat{\eta}_m$ through simple OLS regression which can be easily parallelized across LSVs. The corrected read rates $R^*_{m,k}$ are then computed as:

$$R'_{m,k} = F^{-1}(C(F(\widehat{R_{m,k}}) - N_k \cdot \gamma_m - U_k \cdot \delta_m)); T'_{l,k} = \Sigma_m R'_{m,k} s.t. S(m) = l \qquad (3)$$

$$R^*_{m,k} = \frac{T_{l,k}}{T'_{l,k}} R'_{m,k}$$

Where $C(x) = x$ if $x > 0$ otherwise $C(x) = 0$ is a simple clipping function to avoid too low (negative) read rates after confounders effect removal. We found clipping to affect approximately 7% of the junctions in our tests and it did not seem to have a significant effect on the model accuracy compared to non-clipped junctions (see Supplementary Fig. 6). We also note that the normalization factor $\frac{T_{l,k}}{T'_{l,k}}$ is set to recover the same total level of read rates per LSV i.e. $T^*_{l,k} = T_{l,k}$. The adjusted read $R^*_{m,k}$ rates are reported back as the output for any downstream algorithm.

Finally, a few additional elements of the MOCCASIN algorithm are worth noting:

If an LSV is not observed in a specific sample it is not adjusted.
In order for MOCCASIN to be able to perform a correction the design matrix needs to be full rank.
Even when the design matrix is full rank it might not be full rank for specific LSV due to missing observations (see above). To account for that, MOCCASIN's output includes a list of all the LSV it corrected y/n.
When MOCCASIN is applied to MAJIQ's output, every LSV has multiple read rate estimates for each of its associated junctions. MOCCASIN is then applied to each of those read rates estimates independently, resulting in a set of corrected read rates per junction, per LSV. These sets are combined together by the MAJIQ algorithm to output posterior probabilities for PSI and dPSI as described previously[17].
In order to learn unknown confounders, denoted by the matrix U above, we utilize a procedure similar to the one used by RUV[7] - See supplementary methods for details.

**Reporting summary**. Further information on research design is available in the Nature Research Reporting Summary linked to this article.

## Data availability

The TARGET results published here are in whole or in part based upon data generated by the Therapeutically Applicable Research to Generate Effective Treatments initiative, phs000218. The TARGET data used for this analysis were accessed under Project #10088: Alternative splicing in pediatric cancers (request 41466-5). The SRA toolkit was used to download sra files for the TARGET dataset and simulated data was based off of data from ref.[28] (GEO accession GSE54652). Simulated data generated for this manuscript with and without batch effects are available at GEO accession GSE162664. Mouse RNA-Seq data used for Supplementary Fig. 1 are available at GEO accessions GSE44229 and GSE63412. The ENCODE project fastqs were downloaded from www.encodeproject.org

(See Supplementary Data 1 for a list of file accessions used). Data and scripts to reproduce figures are available in a Zenodo repository (doi:10.5281/zenodo.4294189).

## Code availability

MOCCASIN: https://bitbucket.org/biociphers/moccasin/. Data and scripts to reproduce figures are available in a Zenodo repository (https://doi.org/10.5281/zenodo.4294189).

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

## Acknowledgements

We thank Dr. Manuel Torres Diz for help with the TARGET dataset. This work was supported by R01 GM128096 to Y.B; U01 CA232563 to Y.B., A.T.T, and K.W.L. The results published here are in whole or part based upon data generated by the Therapeutically Applicable Research to Generate Effective Treatments (TARGET) initiative, phs000218, managed by the NCI. The data used for this analysis are available at www.ncbi.nlm.nih.gov/projects/gap/cgi-bin/study.cgi?study_id=phs000218.v22.p8. Information about TARGET can be found at http://ocg.cancer.gov/programs/target.

## Author contributions

Conceptualization: Y.B. and B.S.; Methodology: Y.B and B.S.; Software: B.S. and P.J.; Investigation: Y.B., C.M.R., and B.S.; Formal analysis: Y.B., C.M.R., B.S., and A.J.; Synthetic data generation: N.F.L. and G.R.G.; Data acquisition and curation: C.M.R., A.T., and Y.B.; Visualization: C.M.R. and B.S. Writing, review and editing: Y.B., C.M.R., B.S., K.W.L., G.R.G., N.F.H., and A.T.; Writing, original draft: Y.B. and C.M.R.; Supervision: Y.B. and K.W.L.

## Competing interests

The authors declare no competing interests.
