## [Peer Review File · Nature Communications]

REVIEWER COMMENTS

Reviewer #1 (Remarks to the Author):

In this manuscript, Slaff and Radens et al., developed a new method to assess and correct the effect of confounders in RNA splicing quantification. They found a significant "batch" effect on splicing analysis in two of the more popular datasets publicly available and adjusted the quantification provided by their previous method, MAJIQ, using their new method, MOCCASIN. MOCCASIN employs a linear model to represent PSI values as resulting from factors of biological interest as well as known and unknown confounding factors (not of interest). It then uses the fitted model to adjust PSI to remove the impacts of the confounders.

This new method represents an important tool, as batch effects and technical explanations for differences in isoform usage between conditions are often insufficiently considered in RNA-seq analyses. This is especially problematic, as a wide range of common technical and biological sources of bias are known to confound RNA-seq data. Extending these type of corrections already accomplished in gene expression analyses to splicing quantifications will be broadly useful to the field. However, there are a few specific points that could be developed further.

1. While the presented analyses suggest that the approach is effective, I think that the PSI value adjustment needs to be addressed further. PSI values exist between 0 and 1 by definition, however, after MOCCASIN's adjustment, PSI values can be positive or negative. This requires redefining negative values to equal zero and a renormalization step to ensure that each LSV's PSI values appropriately sum to one. This correction is a post-hoc adjustment that can have significant implications. Can the authors comment on the consequences of this adjustment? Moreover, can the authors compare the adjusted PSI values where this correction was necessary to those where it was not?
2. It is my understanding that MAJIQ employs a generative model that ensures that estimated PSI values are properly constrained to exist between 0 and 1, and that each LSV's PSI values sum to 1. Thus, there are standard statistical frameworks that can be applied such that the predicted variable is constrained, and the authors had used similar approaches in the past. Can the authors handle this constraint in a statistically principled manner? Were, for example, logit models considered?
3. It is unclear to me how the percentages of variance explained by batch labels were calculated in Figure 1 and Figure 3. This is crucial as the whole study is centered on the idea that confounders have a significant effect on splicing analysis. Furthermore, the authors say that "detecting differences between samples (...) is generally maintained ...". If the splicing differences between samples are generally maintained, that would argue that confounders do not have such a significant effect on splicing quantifications. Can the authors explain this further?
4. The manuscript mostly evaluates the effects of batch labels (using the same sequencer or multiple sequencers). Can the analysis be expanded so we understand the effect of other confounders, and specifically unknown confounders, in splicing quantifications? Also, the method is designed to work with the previous method MAJIQ. Can MOCCASIN be expanded to work with other methods as well?
5. Splicing is actually a very broad term. Can the analysis be partitioned into different splicing categories such as skipped exons, intron retention, alternative 3' and 5' splice sites, etc.?

Reviewer #2 (Remarks to the Author):

The Authors present MOCCASIN, a novel methodology to correct for (known and unknown) confounders in alternative splicing analyses, from bulk RNA-seq data. Their approach focuses on percent spliced in (PSI) and on delta PSI (dPSI) to study alternative and

differential splicing, respectively.

The manuscript investigates the effect of confounders (e.g., batch effects) on two real datasets, and the variation (measurement unclear) associated with batches is at least as big for alternative splicing (PSI and dPSI) as it is for transcript expression (TPMs and log-FCs).

MOCCASIN inputs PSI values and returns corrected PSIs; the model assumes PSIs follow an additive model, and corrects PSIs by subtracting the estimated effect of (known and unknown) confounders. Unknown confounders first inferred from RUV (Risso et al., 2014) and then included in MOCCASIN and treated as known confounders.

The Authors show how the effect of confounders is reduced by MOCCASIN in one simulated and two real datasets.

MOCCASIN is released as a python implementation.

I feel the work presented is interesting, covers a rarely studied topic and has good potential to be of interest for a large audience.

However I also think it has to undergo a long list of major edits (apologies if they sound too harsh), as listed below.

Major comments:

1) Concerning the availability of scripts and usage of the method.

1.1) Some scripts are available as Supplementary material: the code used to perform all analyses of the manuscript should be easily accessible online; furthermore, such scripts should also be associated to a DOI (e.g., via zenodo).

Once created, please also report the link to the scripts in the main text.

I also did not like the fact that, an open access method, has methodological details which available only to internal lab members.

1.2) Furthermore, I think MOCCASIN would benefit from clearer install descriptions, and a wiki showing a full example usage (something similar to Bioconductor packages' vignettes).

2) I believe the Introduction should be enhanced with further context.

2.1) Firstly, the authors only illustrate PSIs and dPSIs as a way to investigate alternative and differential splicing.

Indeed, there are several other approaches to alternative and differential splicing; notably, differential transcript usage (DTU, e.g., BANDITS), differential transcript expression (DTE, e.g., cjBitSeq) and differential exon usage (DEU, e.g., DEXSeq).

In the Introduction, I believe it should be acknowledged that there are several ways to quantify alternative splicing (e.g., PSIs, TPMs, relative transcript abundance, exon abundance) and perform differential splicing (e.g., DTU, DTE, DEU and via PSIs).

2.2) Also, SUPPA (2015) is referred, but maybe the more recent SUPPA2 (2018) should be mentioned.

2.3) Furthermore, the Introduction neglects others methods for differential splicing that allow for covariates (such as batch effects); notably, DRIMSeq (Nowicka and Robinson, 2016) and DEXSeq, which can be used on exon counts (Anders et al., 2012), transcript estimated counts (Love et al., 2018) and equivalence classes counts (Cmero et al., 2019).

3) I feel the work lacks a critical assessment of its limitations:

3.1) As far as I understand, local splicing variations (LSVs) are based on junction reads only and ignore other reads; (if my understanding is correct) this should be acknowledged.

3.2) It is also important to explicitly mention that MOCCASIN can only correct PSIs, and not other forms of alternative splicing quantification, such as transcript (or exon) total and relative abundance.

3.3) MOCCASIN seems to only be able to work in conjunction with MAJIQ; however it is not clear if reads can be aligned via multiple tools (e.g., STAR, Salmon, kallisto, etc...). Please put MOCCASIN and MAJIQ in context, by briefly describing a potential usage pipeline (from raw reads), also illustrating what tools could be used to align reads.

3.4) The simulation study relies on strong assumptions (batches up-regulating some transcripts while down-regulating others), see point 5) below.

3.5) The number of parameters (intercept + known and unknown covariates) has to be smaller than the number of samples (connected to point 6 below).

4) I find several parts of the Results Section lack some motivation, details and mathematical rigour:

4.1) the TARGET and ENCODE datasets used are only vaguely introduced; please introduce the data in more detail: references, samples, groups (experimental conditions), etc... 4.2) the fraction of variation explained by batches is unclear: what do these number exactly represent and how are they calculated ?

is this the R^2 calculated by fitting a linear model with batch effect as covariate?

4.3) the additive model for PSI (top page 4) should be better explained and motivated: why is it reasonable to use an additive model on PSIs? What are the assumption of the residuals (e.g., gaussian, 0 mean and homoscedastic)? Did you also consider and test alternative formulations, e.g., an additive model on $\log(\text{PSIs})$?

I find the additive choice unusual, given that PSIs range between 0 and 1 (and force you to "clip" negative values): did you consider a logit (or probit) transformation (i.e., $\text{logit}(\text{PSI}) = \alpha + N \cdot \gamma + U \cdot \delta + V \cdot \eta + \epsilon$)?

Furthermore, all mathematical elements must be introduced immediately after they appear (in this case they are first introduced in the Supplementary material); also the details reported in the Methods lack mathematical rigour: indices "m" and "k" are not defined.

It would be good to also see the mathematical implementation of the OLS algorithm (possibly in Methods or Supplementary).

5) I am sceptical about some aspects the simulation study.

5.1) If my understanding is correct, differential splicing events are not simulate, yet MAJIQ is used to infer differential splicing and these results are set as ground truth ("Running MAJIQ on these data produced the results labeled as ground truth").

This is highly unusual: the main advantage of simulated data is that one "true" (differential) state is known; I encourage you to simulate differential splicing effect, or use some differential splicing simulated data which is already available (e.g., Tiberi and Robinson 2019, or Sonesson et al. 2016), and make full ROC or TPR vs. FDR curves to compare detection of differential splicing, before and after using MOCCASIN.

5.2) The Authors simulate a batch effect by altering the expression (TPMs) of two transcripts, while keeping the other transcripts of the same gene unchanged.

This assumes that batch effect will up-regulate the expression of one/some transcripts while down-regulating one/other transcripts (of the same gene). I wonder how realistic these assumptions are. I think this choice needs to be motivated, by examples in the literature or real data analyses.

Fig 1d), showing how much the "variation" of dPSI is explained by batch effects, is a partial motivation for the simulation design, but I feel there needs to be stronger evidence.

6) The Authors only present a simulation study with 16 samples, 2 groups and 2 batches.

This makes the OLS fit, and estimation of parameters fairly simple: 3 parameters (intercept, batch effect and group) need to be inferred from 16 samples.

I would like to see how robust MOCCASIN is when the number of parameters approaches the number of samples (e.g., with 6 samples, 2 groups and 3 batches).

Minor comments:

1) I think the manuscript never mentions that MOCCASIN is implemented as in python. This should be indicated somewhere.

2) I feel some references are missing or postponed at the end of the sentence; e.g., TARGET and ENCODE datasets should be referenced as soon as they are introduced (page 3). The same goes for MAJIQ (page 4).

3) Page 5: "detecting differences between samples that represent true biological signals is generally maintained".

This is a real data application, where "true biological signals" are unknown (and can only be inferred with error); please re-phrase the sentence, and explain how such signals are calculated.

4) It would be very helpful to have a computational benchmark of the method (with time and RAM usage) on one of the datasets considered.

5) The Authors present results of MOCCASIN using known covariates, while only some plots are reported when the method uses unknown covariates. It would be good to see all results in both cases; specifically MOCCASIN using unknown covariates is missing in Fig 2.e and in all images of Fig 3.

6) typo: fig 2 caption: "Here positive events where considered" -> "Here positive events were considered".

7) The paper states that MOCCASIN also works with bootstrap replicates from MAJIQ, but does not clarify how these are treated.

I guess the PSIs of each bootstrap replicate are corrected and returned to MAJIQ for differential analyses ?

Please briefly clarify in the text.

8) Supplementary Section 2.3 seems redundant; I think it is (almost) identical to the Methods Section.

Simone Tiberi, University of Zurich

We would like to take this opportunity to thank both reviewers for their detailed comments and constructive feedback. As you will find below we have spent considerable time pondering over those comments and working to address concerns raised. We detail our response to each point raised below.

Reviewer #1

While the presented analyses suggest that the approach is effective, I think that the PSI value adjustment needs to be addressed further. PSI values exist between 0 and 1 by definition, however, after MOCCASIN's adjustment, PSI values can be positive or negative. This requires redefining negative values to equal zero and a renormalization step to ensure that each LSV's PSI values appropriately sum to one. This correction is a post-hoc adjustment that can have significant implications. Can the authors comment on the consequences of this adjustment? Moreover, can the authors compare the adjusted PSI values where this correction was necessary to those where it was not?

It is my understanding that MAJIQ employs a generative model that ensures that estimated PSI values are properly constrained to exist between 0 and 1, and that each LSV's PSI values sum to 1. Thus, there are standard statistical frameworks that can be applied such that the predicted variable is constrained, and the authors had used similar approaches in the past. Can the authors handle this constraint in a statistically principled manner? Were, for example, logit models considered?

Reviewer #1 comments above (and similar ones from Reviewer #2 below) raise several significant points we needed to address, regarding our modeling assumptions.

First, we want to clarify that in our description of the method in the original submission we referred to adjusting PSI which led to the above questions about negative PSI. While technically correct because the original model only considered a linear scaling of LSV reads (which can be assumed to add up to 1 and thus represent PSI), in retrospect and given the alternative model we describe below, we realized it is more appropriate to refer to MOCCASIN as modeling scaled read rates. We amended the description as a linear model of scaled read rates so that it now fits both the linear and non linear model.

Both reviewers questioned the validity of a linear model and pointed to a logit transformation as an alternative. Indeed, we originally considered logit as it appears advantageous - commonly used, more principled/elegant, and avoids negative read rate values. However, the logit transformation is problematic for several reasons. First, logit transforms the open interval (0,1) to $[-\infty, \infty]$. Using it does not solve the issue of $\text{psi}=0$ because logit is undefined for 0. We could of course avoid this with some heuristic such as pseudo counts, but it's not clear what those should be. Furthermore, models which utilize this transformation tend to couple it with a homeostatic noise model for simplicity. Such an assumption is highly problematic when the

observed PSI is the result of discrete observations which in most cases are very limited, in the order of 10-20 (junction spanning read counts). This characteristic of splicing quantification is quite different from the realm of gene expression or isoform estimation.

We also explored other alternatives. For example, the DESeq2 approach for modeling covariates as affecting a negative binomial distribution which would capture the read rate per junction. However, this approach was just not scalable for the datasets we intended to use MOCCASIN for.

With all that said, we agree with the reviewers a linear model appears to have clear deficiencies and the response from both reviewers encouraged us to further investigate/explore alternative solutions. Specifically, consider how we model a bias effect in our synthetic data: We identify the most abundant isoform per gene and shift its TPM by X%, while increasing the TPM of another isoform to maintain similar overall gene expression. Such a shift by X%, which can be justified by theoretical models for splicing and empirical observations (see more below), motivates a log linear model. Indeed, the log linear transformation has long been used for gene expression modeling and log is also commonly used as a variance stabilizing transformation. However, we are still left with the issue of 0 read rates. A common solution is to add pseudo counts but we found this to be unstable when we need to scale read coverage and then scale back. A more elegant solution for expression variance stabilization is to use the arcsinh transformation. However, for our purposes arcsinh seems less adequate since its deviation from a log transformation varies as a function of coverage. We thus formulated the following transformation which we term smoothed to zero log transformation (STZL):

$$F(x) = \begin{cases} \ln(x) & x > 2 \\ ax^2 + bx & 0 \leq x \leq 2 \end{cases}$$

$$F^{-1}(x) = \begin{cases} e^x & x \geq \ln(2) \\ \frac{1}{2a} (-b + \sqrt{b^2 + 4ax}) & 0 \leq x < \ln(2) \end{cases}$$

Where:

$$a = \frac{1}{4} (1 - \ln(2))$$

$$b = \ln(2) - \frac{1}{2}$$

STZL solves the issue of deviation from log transformation at high values, gives the desirable $F(x) = 0$ for $x=0$, and the derivative of $F(x)$ equals the derivative of $\ln(x)$ at $x=2$. A plot comparing STZL, log and arcsinh is given below:

Regardless of alternative model choices to consider, the reviewers raised an important question: Can we better quantify the accuracy of MOCCASIN, to assess (a) what is the price associated with the linear assumption and (b) check whether the clipping of read rates to zero causes increased inaccuracies.

First, using our synthetic data (which in turn was derived from real samples) we created new plots that summarize the inaccuracies in PSI correction. The plots below summarize as a CDF the observed differences between PSI computed from the data with batch injection and original data with no batch effect (blue), contrasting it with the CDF of observed differences between PSI computed after MOCCASIN correction (linear model) and the original data. Here we used a total of 4-16 samples (4 samples means only 1 sample per batch/tissue combination, 8 samples means 2 samples per batch/tissue combination etc.) to address Reviewer #2 comment below about assessing MOCCASIN with few samples to infer the bias term (4 is the minimal number as we then have only one sample per tissue/batch combination).

This figure is now included as main Figure 2a. We note that here we perturbed 20% of the genes by 60% TPM of the main transcript so we expect only about 20% of the LSV to show significant deviation in the blue curve, hence the zoom in version of the CDF on the top 20% (other LSV could of course be slightly affected during the data production and PSI quantification, leading to small PSI perturbations). We see that the cumulative frequency (Y-axis) of LSV with dPSI > 20% drops from ~6% (Before MOCCASIN, blue line) to 1-2% (After MOCCASIN, purple/brown/yellow/orange lines). Specifically, the number of LSVs with dPSI > 20% drops from 1229 to 345-151 (purple, brown, yellow, and orange lines for 1, 2, 3, or 4 samples per batch and condition combination) i.e. up to 88% reduction in the number of highly perturbed LSV.

We also performed the same analysis comparing the (MOCCASIN default) linear vs STZL models showing that, for our simulated dataset, the linear and STZL models have similar accuracy:

This figure is now included as Supplementary Figure 8.

We performed similar analysis for comparing the junctions involving negative read rates which were clipped to LSV with junctions without clipping. The results of this analysis are shown in the figure below. Overall, we find across all samples analyzed with the maximum bias factor applied (G=20%, C=60%) that 503-607 LSVs are clipped compared to 6048-7133 which are not, i.e. only ~7% are clipped. The relation between the green (non clipped) and red (clipped) line varies a little between samples. As can be expected overall there seems to be some (slight) advantage in accuracy for the non-clipped (green) line, but the differences are not large, sample dependent, and the clipped events as noted above are a minority. In the most extreme case (top

left, sample SRR 1158525) we find that to achieve the same fraction of events (y-axis at 0.8) “costs” an increase of dPSI inaccuracy (x-axis) by 0.04.

This figure is now included as Supplementary Figure 6.

Together these plots address the reviewer’s concern regarding MOCCASIN’s linear model and better quantification of possible issues such as the read rates clipping.

Notably, these plots indicate STZL has no significant advantage over the linear model. This result appears surprising especially given that the bias simulation used to create those PSI effects fits the log model and not a linear model. To investigate why the linear model appears to perform well we created a toy problem where we did the following:

1. Shift the read rates by X%
2. Perform a linear, arcsinh or STZL correction
3. Measure directly the inaccuracy incurred by the incorrect modeling assumption.

Note that in this setup we know exactly the read rates and the % change. As seen in the results below, because of the scaling and normalization involved, the linear model actually introduces less error compared to a log linear or arcsin model (note a single junction scaling for a log linear model will by definition create no error). This analysis is now included as Supp Fig7.

This analysis indicates that the linear model compares favorably under such perturbations. For example for strong effects of 20% change in TPM the incurred inaccuracies are only 2% and even an extreme 90% change causes a 9% dPSI inaccuracy.

To summarize this key point, we believe we have addressed the reviewers concerns regarding the validity of the MOCCASIN model. While we acknowledge alternatives exists, and may form the base for future works, we assess the effectiveness of both the linear model and the potential inaccuracies incurred by clipping negative read rates. Furthermore, we introduced an alternative model using the STZL transformation. While STZL did not appear to offer significant advantages in our testing we include it in the revised manuscript for completion and since its theoretical properties might translate to improved performance in some user/bias specific settings we did not explore. Finally, we significantly extended the discussion of alternative models and limitations in the revised manuscript.

It is unclear to me how the percentages of variance explained by batch labels were calculated in Figure 1 and Figure 3.

The description was indeed unclear. As Reviewer #2 noted, this was simply the R^2 calculated by fitting a linear model with batch effect as a covariate. We revised the text.

The authors say that “detecting differences between samples (...) is generally maintained ...”. If the splicing differences between samples are generally maintained, that would argue that confounders do not have such a significant effect on splicing quantifications. Can the authors explain this further?

We think this is a misunderstanding due to a lack of clarity of our original writing. What this paragraph meant to say is that after MOCCASIN was applied, the % of the overall variance

associated with the batch effect decreased while the % of overall variance associated with a known biological signal increased.

Can the analysis be expanded so we understand the effect of other confounders, and specifically unknown confounders, in splicing quantifications?

The reviewer raises an important question here, which is also reflected in Reviewer #2 comments. To address it, we expanded our analysis using unknown confounders for synthetic data (Fig S2, S3) and for the TARGET dataset. TARGET is an interesting test case given the size but also complexity of the data: heterogeneous samples, multiple centers, different technologies, conditions, and cell types. In the revised manuscript we show how this analysis first identified a key confounder we originally missed in the annotation (cell type), and how it can be used to explore additional sources of variations remaining in the data. In the future, we believe analysis as we performed here would form the base for future exploration and usage of MOCCASIN on large heterogeneous datasets to gain better understanding of sources of variations in different datasets.

Can MOCCASIN be expanded to work with other [non-MAJIQ] methods as well?

Yes. In response to both reviewers' requests we spent much effort revising MOCCASIN's code. We implemented a general API such that MOCCASIN can work with splicing quantification methods other than MAJIQ. This ability is now clearly stated in the revised manuscript.

Splicing is actually a very broad term. Can the analysis be partitioned into different splicing categories such as skipped exons, intron retention, alternative 3' and 5' splice sites, etc.?

Indeed, we very much agree with the reviewer that differences between alternative splicing event types when it comes to confounders' effect is a very interesting one to pursue. Since little is known about this, doing it justice will require dissecting events by types across different datasets and confounders types, then trying to assess the reasons and implications of any observed differences. Indeed, we have another publication in the pipeline which introduces MAJIQ V2 that will enable researchers to ask such questions more easily and systematically. We believe properly addressing this question is well beyond the scope of this paper which is focused on introducing MOCCASIN to handle confounders in splicing data. However, since this is indeed a very interesting question we added a segment in the discussion to raise it as a future direction of investigation.

Reviewer #2

Some scripts are available as Supplementary material: the code used to perform all analyses of the manuscript should be easily accessible online; furthermore, such scripts should also be associated to a DOI (e.g., via zenodo). Once created, please also report the link to the scripts in the main text.

I also did not like the fact that, an open access method, has methodological details which available only to internal lab members.

The link to Google Doc in the Repository was in fact a draft version of the supplemental methods document. The supplemental document contains all the methodological details.

I think MOCCASIN would benefit from clearer install descriptions, and a wiki showing a full example usage (something similar to Bioconductor packages' vignettes).

We thank the reviewer for these suggestions. To address these, we included an example dataset in the Zenodo repository including a README on how to run MOCCASIN on the example dataset. We also included an example downstream analysis pipeline to show how one can visualize and quantify the impact of batch effects on splicing quantification using the example dataset before and after running MOCCASIN. The link to this example dataset is included in the MOCCASIN repository README, which also includes a brief methods description, detailed information on how to configure the input model matrix for MOCCASIN, and examples of how MOCCASIN can be invoked to adjust for confounding factors.

I believe the Introduction should be enhanced with further context. Firstly, the authors only illustrate PSIs and dPSIs as a way to investigate alternative and differential splicing. Indeed, there are several other approaches to alternative and differential splicing; notably, differential transcript usage (DTU, e.g., BANDITS), differential transcript expression (DTE, e.g., cjBitSeq) and differential exon usage (DEU, e.g., DEXSeq). In the Introduction, I believe it should be acknowledged that there are several ways to quantify alternative splicing (e.g., PSIs, TPMs, relative transcript abundance, exon abundance) and perform differential splicing (e.g., DTU, DTE, DEU and via PSIs).

Following this we extended the discussion describing various approaches to alternative splicing detection/quantification before delving into PSI/dPSI which are the focus of this paper. We note though that much of the work cited by Reviewer #2 above and below focuses on differential splicing analysis (with the ability to correct for confounders). This is **not** the focus of this work, which is about correcting PSI estimates. Correcting PSI estimates can be used for differential splicing analysis (which is some of the use cases we demonstrate) but also other types of analysis, of which clustering (unsupervised) in PSI space is a prime example which we demonstrate as well. We included a description in the main text to make this point clear.

Also, SUPPA (2015) is referred, but maybe the more recent SUPPA2 (2018) should be mentioned.

Fixed.

Furthermore, the Introduction neglects others methods for differential splicing that allow for covariates (such as batch effects); notably, DRIMSeq (Nowicka and Robinson, 2016) and DEXSeq, which can be used on exon counts (Anders et al., 2012), transcript estimated counts (Love et al., 2018) and equivalence classes counts (Cmero et al., 2019).

We now note that methods that use transcript expression estimates can have their output corrected for confounders using existing methods. Please also see note above about the distinction we draw between methods for differential splicing analysis and correcting PSI estimates.

As far as I understand, local splicing variations (LSVs) are based on junction reads only and ignore other reads; (if my understanding is correct) this should be acknowledged.

It is also important to explicitly mention that MOCCASIN can only correct PSIs, and not other forms of alternative splicing quantification, such as transcript (or exon) total and relative abundance.

We now make it clear in the text that LSVs are based on junction spanning reads (and reads across introns for IR events) and that MOCCASIN aims to correct PSI and not transcript/exon expression/abundance.

MOCCASIN seems to only be able to work in conjunction with MAJIQ; however it is not clear if reads can be aligned via multiple tools (e.g., STAR, Salmon, kallisto, etc...). Please put MOCCASIN and MAJIQ in context, by briefly describing a potential usage pipeline (from raw reads), also illustrating what tools could be used to align reads.

As noted in the response to Reviewer #1 above, we have revised MOCCASIN to implement an API such that other tools for quantifying PSI can be used with it.

The number of parameters (intercept + known and unknown covariates) has to be smaller than the number of samples (connected to point 6 below).

We agree with Reviewer #2 this is a limitation. More specifically, the design matrix needs to be full rank. Furthermore, unlike expression data where unobserved values can usually be assumed to be 0, this is not the case for PSI quantifications and missing values are more common. This means that unless some imputation is performed even if the design matrix is full rank for fully observed LSV, some rows (LSV) may not have full rank due to missing values. In such cases the LSV will not be adjusted. MOCCASIN outputs a report that lists which LSV have not been corrected. We include the above in the discussion section where we highlight limitations and future directions.

The TARGET and ENCODE datasets used are only vaguely introduced; please introduce the data in more detail: references, samples, groups (experimental conditions), etc...

We now include more detailed description of these datasets.

The fraction of variation explained by batches is unclear: what do these number exactly represent and how are they calculated ? Is this the R^2 calculated by fitting a linear model with batch effect as covariate?

This point was raised by Reviewer #1 as well (see above). Indeed the values reported are just the R^2 for batch effect as a covariate.

Why is it reasonable to use an additive model on PSIs?

Reviewer #2 raises here a fundamental question regarding modeling of effectors. This question goes well beyond splicing to general factors/signals affecting gene expression and is still under considerable debate/investigation (see for example a recent exploration of this question in Sanford et al. eLife 2020). In the case of splicing, some theoretical support for how factors affect splicing outcome given a specific splicing context (a specific event, in a specific condition) can be found in the recent work by (Baeza-Centurion et al, Cell, 2019). Briefly, the authors show that the following model fits well with several data: The probability that a splice site is recognized by the splicing machinery remains fixed with some constant, such that the probability of a given exon to be included at a specific time t follows an exponential. PSI is a result of the competition between two (or more) competing splice junctions, each with its own constant (and time delay between them). The effectors, genetic variants in the case of this work by Baeza-Centurion et al, affect the constant associated with the matching splice junction. Thus, this model points to a linear combination of effectors given a specific context (event, condition) in log space for the matching junction spanning reads.

The above model supports the synthetic modeling of effects on splicing where we increase the expression of the affected isoform (associated with its splice junctions) by $X\%$ (additive in log space). However, we showed both on real and synthetic data (which notably does not fit the MOCCASIN model assumption) that the original linear model performed well and using a toy dataset illustrated how this somewhat counter intuitive result may be explained for PSI modeling (see above response to Reviewer #1).

In addition, previous works have shown that when comparing two conditions most of the splicing changes at the local events involve binary like changes even when the event itself is complex (i.e. involves multiple junctions) i.e. a highly included junction goes down while another one goes up. Thus, a confounder (known/unknown) which is the result of such a signal (e.g. mixture of cell types/tissue contamination) would fit our synthetic modeling approach for reducing the major isoform by $X\%$ while increasing another isoform by the same TPM to maintain similar expression levels. Still, this “binary like” behaviour is definitely not always the case.

Furthermore, the above models/observations were not derived specifically for batch effects.

Thus, we would be the first to agree that the question of how are splicing variations distributed

due confounders is a fascinating one. We are also not aware of any previous work that addresses this question. But as we noted regarding a related question posed by Reviewer #1 about the distribution of event types, addressing this question requires extensive evaluation across datasets and confounder types which is well beyond the scope of this work.

In summary, we believe both previous literature and our empirical results give sufficient support for our modeling assumptions. That said, we very much agree with Reviewer #2 that such a detailed discussion about our modeling assumptions/limitations was missing in the original submission and we added it to several parts of the main text in the revised manuscript.

What are the assumptions of the residuals (e.g., gaussian, 0 mean and homoscedastic)?

We use an L2 penalty which implies residuals for junction's read rate are distributed as a Gaussian, with 0 mean and homoscedastic.

Did you also consider and test alternative formulations, e.g., an additive model on $\log(\text{PSIs})$? I find the additive choice unusual, given that PSIs range between 0 and 1 (and force you to “clip” negative values): did you consider a logit (or probit) transformation (i.e., $\text{logit}(\text{PSI}) = \alpha + N\gamma + U\delta + V\eta + \epsilon$)?

Please see response above to a similar comment by Reviewer #1 and the STZL model we introduced in the revision.

Furthermore, all mathematical elements must be introduced immediately after they appear (in this case they are first introduced in the Supplementary material); also the details reported in the Methods lack mathematical rigour: indices “m” and “k” are not defined.

It would be good to also see the mathematical implementation of the OLS algorithm (possibly in Methods or Supplementary).

We apologize for the lack of details and clarity. We worked to address mathematical clarity/rigour in the revision. We are not sure what is meant by including the OLS implementation but all code is available. Specifically, OLS code can be found in

https://bitbucket.org/biociphers/scf_tools/src/971909612745fd833b65d5a9f389334306509420/moccasin/fit_adjust.py

#lines-451,454,461 and this corresponds to the usual mathematical implementation of OLS (e.g. https://en.wikipedia.org/wiki/Ordinary_least_squares#Estimation)

The Authors simulate a batch effect by altering the expression (TPMs) of two transcripts, while keeping the other transcripts of the same gene unchanged. This assumes that batch effect will up-regulate the expression of one/some transcripts while down-regulating one/other transcripts (of the same gene). I wonder how realistic these assumptions are. I think this choice needs to

be motivated, by examples in the literature or real data analyses. Fig 1d), showing how much the “variation” of dPSI is explained by batch effects, is a partial motivation for the simulation design, but I feel there needs to be stronger evidence.

Please see the discussion above which includes justification based on previous research to the approach we took to model confounders. In addition, to address the concern about measuring more directly the observed effects we included new analysis which measures how well the correction works compared to the unaffected data as a CDF over deviation in PSI (measured as dPSI) which captures directly the error in corrections (see response to Reviewer #1 above).

If my understanding is correct, differential splicing events are not simulated, yet MAJIQ is used to infer differential splicing and these results are set as ground truth (“Running MAJIQ on these data produced the results labeled as ground truth”). This is highly unusual: the main advantage of simulated data is that one “true” (differential) state is known; I encourage you to simulate differential splicing effect, or use some differential splicing simulated data which is already available (e.g., Tiberi and Robinson 2019, or Sonesson et al. 2016), and make full ROC or TPR vs. FDR curves to compare detection of differential splicing, before and after using MOCCASIN.

We respectfully (partially) disagree here with Reviewer #2. First, to clarify some possible misunderstanding, the data *is* simulated and not by MAJIQ. We take real samples, use a separate software, RSEM, to compute TPM per isoform, then simulate reads based on those. Thus, the “true” data is known, and MAJIQ is not used to set up the dataset based on its own internal model. Notice, however, that unlike the works listed above, the purpose of this work is *not* to evaluate MAJIQ but MOCCASIN. Thus, we think it makes sense to directly compare MAJIQ’s output before/after MOCCASIN (Otherwise inaccuracies would be the combination of both). Accordingly, instead of ROC, TPR, FDR etc. which makes sense for assessing differential splicing methods as in the works mentioned, we evaluate both the quality of corrections (new CDF of dPSI compared to the data without a batch effect described above), and how the corrections effect MAJIQ’s calls of differential splicing (with matching assessment of FDR etc. before/after).

Another point to make is that the works listed by Reviewer #2 simulate not only the differential splicing signal but also the expression and the variation between samples belonging to the same group. Specifically, they fit a NB model and then sample from it to create replicates. This is a very strong assumption which in our experience does not hold for splicing quantifications from heterogeneous groups such as patients vs. controls or GTEX donors. We avoid assumptions on expression and variations within the group by modeling the samples based on real samples. This means that variability within each group, and the scope of expression observed in real samples, is more likely to be maintained/realistic. Thus, while we agree with Reviewer #2 the approach taken by these works gives a very controlled environment for testing, we argue that it also suffers from additional assumptions which are likely unrealistic.

In summary, we definitely agree with Reviewer #2 other approaches for simulating data have merit and can carry specific benefits. But we also argue our synthetic evaluation procedures are very reasonable given the specific task at hand. We also tried to make the above considerations more clear in the revision so readers can have a more clear understanding of our modeling assumptions.

The Authors only present a simulation study with 16 samples, 2 groups and 2 batches. This makes the OLS fit, and estimation of parameters fairly simple: 3 parameters (intercept, batch effect and group) need to be inferred from 16 samples. I would like to see how robust MOCCASIN is when the number of parameters approaches the number of samples (e.g., with 6 samples, 2 groups and 3 batches).

This is a good point and a valid concern we did not address in our original manuscript. We now added evaluations of performance with a total of 4,8,12, to the original evaluation with 16 samples. 4 samples is the minimal possible (1 sample per batch/condition combination). We also note that the ENCODE data includes much more than 2 known batches.

I think the manuscript never mentions that MOCCASIN is implemented as in python. This should be indicated somewhere.

Fixed.

I feel some references are missing or postponed at the end of the sentence; e.g., TARGET and ENCODE datasets should be referenced as soon as they are introduced (page 3). The same goes for MAJIQ (page 4).

Fixed.

Page 5: “detecting differences between samples that represent true biological signals is generally maintained”. This is a real data application, where “true biological signals” are unknown (and can only be inferred with error); please re-phrase the sentence, and explain how such signals are calculated.

It would be very helpful to have a computational benchmark of the method (with time and RAM usage) on one of the datasets considered.

We now include information for both memory and time on the 4-16 synthetic samples that model a real dataset, as well time/mem for TARGET (885 samples). See also the figure below. The main bottleneck is I/O (writing back files) when processing large datasets.

The Authors present results of MOCCASIN using known covariates, while only some plots are reported when the method uses unknown covariates. It would be good to see all results in both cases; specifically MOCCASIN using unknown covariates is missing in Fig 2.e and in all images of Fig 3.

We include Supp Fig S2, S3 with the equivalent for Fig 2e using unknown covariates. We added evaluation of MOCCASIN with unknown covariates for TARGET.

typo: fig 2 caption: “Here positive events where considered” -> “Here positive events were considered”.

Fixed.

The paper states that MOCCASIN also works with bootstrap replicates from MAJIQ, but does not clarify how these are treated. I guess the PSIs of each bootstrap replicate are corrected and returned to MAJIQ for differential analyses ? Please briefly clarify in the text.

Good point - we added a description for this.

Supplementary Section 2.3 seems redundant; I think it is (almost) identical to the Methods Section.

We revised the manuscript so the description of the method is only included once. We removed redundancy between the supplementary file and the methods section in the main text.

REVIEWERS' COMMENTS

Reviewer #1 (Remarks to the Author):

The authors responded to all my concerns and the new version of the manuscript has significantly improved. I support its publication in Nature Communications.
I would also be excited to see the new MAJIQ V2 pipeline out soon, but would encourage the authors to work on expanding MOCCASIN to handle confounders with different splicing categories.

Reviewer #2 (Remarks to the Author):

The Authors have addressed (or sensibly replied to) most of my concerns, and provided a much improved version of their work.

A few (minor) points are still need to be addresses:

- Supp Fig S2-S3 miss the colours in the legend;
- Fig 2a, please indicate the number of samples in the legend.
Figures should be self-explanatory, without need to look at the text.
Also, what is the "Before MOCCASIN" based on? Shouldn't there be one "before MOCCASIN" for each sample size?
I would have appreciated seeing performance on the various sample sizes (4,8,12,16) on all panels of Fig 2 (eventually as Supp), rather than in 2a alone.
- MAJIQ still lacks a reference (page 4); ENCODE's reference at page 4 should be: De Souza, 2012.
- The Fig including Computational Cost (Time and Memory) is useful!
Please add it to the test or Supplementary (unless I missed, I think it's absent).
- You said you "added a description for this", but I didn't find it; could you point me where?
The word "bootstrap" doesn't appear to be in the text anymore.
- "Fig. 2f shows that the original data clusters by tissues (left), by batch when the batch signal was injected (middle, $G=20$ and $C=60$), and again by tissue after MOCCASIN is applied (right)."
I appreciate Figs S2, S3 and 2f; I think many readers will struggle to interpret these results (i.e., how MOCCASIN decreases the impact of the batch on clustering).
I think more explanation would be beneficial.
- typo? "hemostatic gaussian noise" at page 12.
- "Automatic citation updates are disabled. To see the bibliography, click Refresh in the Zotero tab. "
Please report citations in the text, without requiring reviewers to install/use external tools (personally I use latex and never used Zotero).

A final comment: replies do not indicate where the edits were introduced, and I struggled to find for your amendments!

Replies should have a precise indication of where the text was edited; e.g., "page xx, lines yy".
Highlighting the new sentences (e.g., in yellow) in the text helps a lot (usually journals allows such a version for reviewers).

Also, I don't mind if you change the order of replies, but please keep the numbers associated to reviewers' questions.

Please keep in mind that I/we read your original manuscript months ago...I/we don't exactly remember where things are.

Same goes for shared replies between reviewers: either copy the reply or indicate exactly where I should look for it!

e.g., "Please see response above to a similar comment by Reviewer #1 and the STZL model we introduced in the revision."

Simone Tiberi, University of Zurich

SUBMISSION INFORMATION

We thank both reviewers for their effort and additional comments on our revised manuscript. It took us considerable time (and much more than we hoped) to prepare the manuscript for this 2nd revision since some of the authors started new jobs yet we wanted to make sure comments are addressed properly. We also wanted to make sure the revision is organized appropriately (see Reviewer #2 last comment about previous revision). In addition to the above, we also needed to gain specific NIH permission to include the TARGET data analysis presented here.

We include a point-by-point response to all comments raised. We hope you find this version suitable for publication.

REVIEWERS' COMMENTS

Reviewer #1 (Remarks to the Author):

The authors responded to all my concerns and the new version of the manuscript has significantly improved. I support its publication in Nature Communications. I would also be excited to see the new MAJIQ V2 pipeline out soon, but would encourage the authors to work on expanding MOCCASIN to handle confounders with different splicing categories.

We thank Reviewer #1 for the encouragement and favorable view of the revised manuscript. We very much look forward to having MAJIQ V2 finally out and applied together with MOCCASIN to confounders in different splicing categories.

Reviewer #2 (Remarks to the Author):

The Authors have addressed (or sensibly replied to) most of my concerns, and provided a much improved version of their work.

A few (minor) points are still need to be addresses:

- Supp Fig S2-S3 miss the colours in the legend;

We scanned all figures in both main text and supplementary figures and made sure to define all colors. Specifically, the red lines in these figures dendrograms are now described. In addition, we found that Fig S1a,b the colors in the rightmost panels were not defined. The colors in those panels simply represented the ID of each sample and as such were not informative/relevant for the batch definition/assessment. We therefore simply removed these non informatives sub-panels.

- Fig 2a, please indicate the number of samples in the legend.

Added.

Figures should be self-explanatory, without need to look at the text.

Also, what is the “Before MOCCASIN” based on? Shouldn’t there be one “before MOCCASIN” for each sample size?

I would have appreciated seeing performance on the various sample sizes (4,8,12,16) on all panels of Fig 2 (eventually as Supp), rather than in 2a alone.

We apologize for not making this analysis clear enough. We clarified the labels and the sample size associated with these panels. Specifically, we note Fig2b/c use 4 samples per batch/tissue combination - same as in Fig2a orange line. Fig2a shows accuracy remains stable for 1-4 samples per such group, but we appreciate the Reviewer’s advice to test MOCCASIN performance on a smaller subset of samples for Fig2b-c, so we have added this analysis as Supplementary Figure 2.

- MAJIQ still lacks a reference (page 4); ENCODE’s reference at page 4 should be: De Souza, 2012.

Fixed.

- The Fig including Computational Cost (Time and Memory) is useful!

Please add it to the test or Supplementary (unless I missed, I think it’s absent).

We are glad the Reviewer found it useful. We added it now as a supplementary figure.

- You said you “added a description for this”, but I didn’t find it; could you point me where? The word “bootstrap” doesn’t appear to be in the text anymore.

We apologize for not making this more clear. This is also a result of not properly annotating and organizing the response as the reviewer noted below. We apologize for this.

We explain in the revised text that MAJIQ outputs multiple estimates of read rates from each LSV’s junctions, which in turn led to multiple corrected “versions” of the read rates. MAJIQ can take advantage of these to create a posterior distribution over PSI/dPSI estimates. We include below the excerpt from the main text:

“Beyond the modeling assumptions, there is also inherent limitation in outputting a single “corrected” version of the original RNA-Seq data. Such output by itself does not include information about the credible interval for the underlying estimated parameters for confounders’ effect, and consequent possible variability in the “corrected” output. This issue has already been noted for methods correcting gene expression and downstream analysis such as differential

expression (Nygaard, Rødland, and Hovig 2016). In the case of MOCCASIN, the model enables users to specify multiple estimates for the read rates per junction, which results in multiple corrected “versions” of the read rates. This feature enables MOCCASIN to take advantage of MAJIQ’s model which outputs multiple estimates for read rates per junction, and then combines those into posterior probabilities for PSI and dPSI estimates. However, while this MAJIQ feature was designed to accommodate for the uncertainty in PSI estimates it does not directly model uncertainty in confounders coefficients, leaving the modeling of such uncertainty as another future direction to improve upon MOCCASIN’s model.”

- “Fig. 2f shows that the original data clusters by tissues (left), by batch when the batch signal was injected (middle, $G=20$ and $C=60$), and again by tissue after MOCCASIN is applied (right).”

I appreciate Figs S2, S3 and 2f; I think many readers will struggle to interpret these results (i.e., how MOCCASIN decreases the impact of the batch on clustering).

I think more explanation would be beneficial.

We are glad the reviewer found these figures useful. We improved the explanation of what is “Ground truth”/“Before”/“After” (see above) so we hope these make interpretations easier. While one could add numerous quantitative measures (FM Index, mutual information, length of tree branches in the dendrogram) we think the result is striking by itself (“a picture is worth a thousand words”) and opted to leave those out for simplicity/clarity.

- typo? “hemostatic gaussian noise“ at page 12.

Indeed! The term is “homoscedastic”. We thank the reviewer for noticing this typo.

- “Automatic citation updates are disabled. To see the bibliography, click Refresh in the Zotero tab. “

Please report citations in the text, without requiring reviewers to install/use external tools (personally I use latex and never used Zotero).

We deeply apologize for this error in the submitted file. No Zotero installation was meant to be required - this was an error message appended to the file since the Zotero directory was not associated with the account from which the submission was performed. We failed to notice the error message as we performed the last round of updates before uploading the file. All bibliography should appear correctly now.

A final comment: replies do not indicate where the edits were introduced, and I struggled to find for your amendments!

Replies should have a precise indication of where the text was edited; e.g., “page xx, lines yy”.

Highlighting the new sentences (e.g., in yellow) in the text helps a lot (usually journals allow such a version for reviewers).

Also, I don't mind if you change the order of replies, but please keep the numbers associated to reviewers' questions.

Please keep in mind that I/we read your original manuscript months ago...I/we don't exactly remember where things are.

Same goes for shared replies between reviewers: either copy the reply or indicate exactly where I should look for it!

e.g., "Please see response above to a similar comment by Reviewer #1 and the STZL model we introduced in the revision."

We could not agree more and again find ourselves deeply apologizing for not meeting our own standards in making the submitted files easy to follow for reviewers. We ran into several challenges during the preparation of the manuscript but these should not have resulted in a harder review task. Please accept our apologies for that. We have made an effort to improve the organization in this submission.